# The Sex Determination Cascade in the Silkworm

**DOI:** 10.3390/genes12020315

**Published:** 2021-02-23

**Authors:** Xu Yang, Kai Chen, Yaohui Wang, Dehong Yang, Yongping Huang

**Affiliations:** 1Key Laboratory of Insect Developmental and Evolutionary Biology, Center for Excellence in Molecular Plant Sciences, Shanghai Institute of Plant Physiology and Ecology, Chinese Academy of Sciences, Shanghai 200032, China; yangxu@cemps.ac.cn (X.Y.); chenkai2015@cemps.ac.cn (K.C.); yhwang@cemps.ac.cn (Y.W.); yangdehong@cemps.ac.cn (D.Y.); 2University of Chinese Academy of Sciences, Beijing 100049, China

**Keywords:** sex determination, cascade, Lepidoptera insects, silkworm

## Abstract

In insects, sex determination pathways involve three levels of master regulators: primary signals, which determine the sex; executors, which control sex-specific differentiation of tissues and organs; and transducers, which link the primary signals to the executors. The primary signals differ widely among insect species. In Diptera alone, several unrelated primary sex determiners have been identified. However, the *doublesex* (*dsx*) gene is highly conserved as the executor component across multiple insect orders. The transducer level shows an intermediate level of conservation. In many, but not all examined insects, a key transducer role is performed by *transformer* (*tra*), which controls sex-specific splicing of *dsx*. In Lepidoptera, studies of sex determination have focused on the lepidopteran model species *Bombyx mori* (the silkworm). In *B. mori,* the primary signal of sex determination cascade starts from *Fem*, a female-specific PIWI-interacting RNA, and its targeting gene *Masc*, which is apparently specific to and conserved among Lepidoptera. *Tra* has not been found in Lepidoptera. Instead, the *B. mori* PSI protein binds directly to *dsx* pre-mRNA and regulates its alternative splicing to produce male- and female-specific transcripts. Despite this basic understanding of the molecular mechanisms underlying sex determination, the links among the primary signals, transducers and executors remain largely unknown in Lepidoptera. In this review, we focus on the latest findings regarding the functions and working mechanisms of genes involved in feminization and masculinization in Lepidoptera and discuss directions for future research of sex determination in the silkworm.

## 1. Introduction

Sexual reproduction is an evolutionarily ancient feature. Recombination of chromosomes from two parents can result in the emergence or the loss of genes during meiosis. Sexes are the prerequisite for sexual reproduction, and sex determination systems are diverse [1]. Different types of sex chromosomes and various sex-determining genes have developed over the course of evolution. In insects, different compositions of sex chromosomes, such as XY, X0, WZ and Z0, are observed, and the signaling pathways involved in sex determination are poorly conserved [2,3,4]. In the model organism *Drosophila melanogaster,* the primary signal of sex determination is composed by X-linked signal elements (XSE) and regulates the transcripts of *Sex-lethal* (*Sxl*), the transducer is encoded by *transformer* (*tra*) and *transformer 2* (*tra2*), and the executors are encoded by *doublesex* (*dsx*) and *fruitless* (*fru*) [5,6,7]. Studies of *Drosophila* and other insects, such as *Apis mellifera, Nasonia vitripennis* and *Tribolium castaneum,* have established that the primary signals coming from the sex chromosomes are diverse among species; whereas the transducers and executors, encoded by genes located on autosomes, are more highly conserved [8,9,10,11,12,13]. The novel sex determination factors that function as primary signals have been identified in Diptera in recent years. They include *Nix*, which is a distant homolog of *tra2* and which was identified in *Aedes aegypti*; *YoB*, which is a maleness gene in *Anopheles gambiae*; *Mdmd*, a splicing factor that was identified in *Musca domestica*; and *MoY*, the Y-linked, male-determining factor in the *Ceratitis capitata*, which is also functionally conserved in *Bactrocera dorsalis, Bactrocera oleae* and many other Teprhtidae species [14,15,16,17,18]. Although these factors are diverse, the primary signal mediated by these genes is transduced by a pathway that leads to alternative splicing of *dsx*, which controls sex differentiation eventually [19]. However, due to the absence of *tra* in mosquitos and because *Sxl* orthologs are not involved in primary sex determination in Medfly and Musca species, only the terminal genes within the sex determination pathway are conserved in Diptera [20]. 

Lepidoptera is a diverse order of insects that includes both pests and species of economic importance such as pollinators and silk producers [21]. The mechanism of sex determination in lepidopterans is quite different from that in dipterans [22]. The lepidopteran model species *Bombyx mori* uses a WZ/ZZ sex determination system, in which homozygous ZZ makes males, whereas heterozygous WZ makes females. This implies that the sex determiner is a female factor (F-factor) from the W chromosome [23]. Studies using high-throughput sequencing and genome editing technologies have identified unique sex determination genes in lepidopterans such as *Silkworm-PIWI* (*Siwi*), *gametocyte-specific factor 1* (*Gtsf1*), *Masculinizer* (*Masc*) and *P-element somatic inhibitor (PSI*), and have preliminarily established the molecular mechanism of the sex determination cascade in Lepidoptera [21,24,25,26,27,28] (Figure 1). Here we summarize the progresses of sex determination studies achieved in recent years in lepidopteran and compare them with the model *D. melanogaster*. We also suggest further directions in the Lepidoptera model *B. mori* that will provide insights into sex determination in other Lepidoptera species. 

## 2. Overview of the Sex Determination Cascade in the Silkworm

The sex determination pathways in insects are diverse [2]. The diversity arises mostly from primary signals, whereas transducers and executors are more conserved [29,30]. In *D. melanogaster*, *M. domestica* and *C. capitata* the primary sex determiner controls sex development via Tra-transduced signaling pathway, resulting in the alternative splicing of *dsx* [5,16,17]. The transducer gene usually encodes a splicing factor, and manipulation by the primary signals therefore exists or functions as a sex determiner only in one sex. The products of the executor gene *dsx* act downstream of these switches. There are female- and male-specific isoforms of DSX, which control sexual development in most insect species [31]. However, in the silkworm, the ortholog of *tra* does not exist, and no *dsx* cis-regulatory element binding sites are found, which indicates the sex determination pathways are markedly different from those seen in flies [20,32]. 

In the lepidopteran model species *B. mori*, the primary signal is the PIWI-interacting RNA (piRNA) *Fem*, which is derived from the W chromosome [24]. *Fem* silences a gene unique to Lepidoptera, *Masc*, which is essential for both male determination and repression of the vital process of dosage compensation of Z-linked genes during embryogenesis (Figure 1) [21,26,33,34,35]. In recent years, sex-determined factors encoded by *PSI*, *Znf-2, Siwi* and *Gtsf1* have been identified along with preliminary established genetic cascade of the sex determination in the silkworm [25,27,28,33,36]. 

## 3. Feminizers, the Primary Signals

Research on sex determination factors in the silkworm began in 1933 when Hasimoto hypothesized that the F-factor originates from the W chromosome [39]. Subsequent genetic studies showed that one copy of the W chromosome is sufficient to determine femaleness [40]. However, in part due to the various transposable elements, their remnants, and simple repeats on W chromosome, the identity of the F-factor was a mystery [41,42]. Only recently was the F-factor identified when researchers used new generation sequencing technologies, bioinformatics and focused on non-coding RNAs (such as microRNAs, long non-coding RNAs and piRNAs). The F-factor is a single piRNA derived from a piRNA precursor named *Fem*, which is transcribed from the W chromosome. *Fem* functions as the primary determinant of female sex in the silkworm [24,43,44,45,46,47].

piRNAs are a class of small RNAs of 24–31 nucleotides in size. They are produced from transposons and from discrete genomic loci called piRNA clusters. The piRNAs guide PIWI proteins to target transcripts [48,49]. In flies, piRNAs and PIWI proteins mainly function in germ cells during gametogenesis to suppress transposable elements, which are selfish genomic elements that are able to jump around the genome [48,49]. It is fascinating that the *Fem*-derived piRNA participates in sex determination, which is a somatic cell fate event in the silkworm. The piRNA pathway is not well characterized in vivo, and as a result how *Fem* functions at the molecular level is still a puzzle [50]. piRNAs have no enzyme activity, and instead they assemble into piRNA-induced silencing complexes (piRISCs) with the PIWI proteins such as Siwi and Argonaute RISC Catalytic Component 3 (Ago3) identified in the silkworm [43]. After loading onto PIWI proteins, piRNAs are produced by a unique biogenesis pathway called the “ping-pong” cycle. The ping-pong cycle is a posttranscriptional gene-silencing mechanism in which RNAs degraded by piRNA-guided transcript silencing provide substrates for additional piRNA production [51]. We have discovered that SIWI is crucial for feminizing the silkworm, whereas Ago3 mutants display no phenotype involved in sex determination [25]. Our studies suggest that SIWI is dominant during *Masc* mRNA silencing via *Fem*-piRNA, whereas Ago3 have minor effects on *Fem* piRNA processing. 

The demonstration that piRNAs have a function in sex determination in the silkworm prompted us try to understand the intricate biogenesis of PIWI-interacting RNAs. The sex determination cascades, as well as piRNA pathways, are distinct in *D. melanogaster* and *B. mori* [50]. The piRNA biogenesis pathway consists of a list of components which are specialized for their processing [52]. However, several key elements of the piRNA pathway in the fly, such as Yb, Rhino, Deadlock and Cutoff, which are crucial for piRNA transcription initiation and piRNA processing, are absent in the silkworm [52]. We have reported that a conserved component of the piRNA pathway called Gtsf1 is involved in female sex determination in the silkworm [28]. In *Drosophila*, Gtsf1 is required for female fertility and interacts with PIWI via its C-terminal end; and it is also essential for piRISCs-induced transposon silencing but not for piRNA biogenesis [53,54]. In the silkworm Gtsf1 is not only necessary for transposon silencing but also for piRNA biogenesis [28]. In addition, our co-immunoprecipitation experiments suggested that Gtsf1 interacts with SIWI. Such an interaction was also observed in *Drosophila* [28,53,54]. Interestingly, not every component from the piRNA pathway participates in feminization. For instance, Maelstrom (Mael) is essential for spermatogenesis and oocyte development in *Drosophila* as it is involved in piRNA-mediated silencing of transposable elements [55]. However, depletion of *Mael* in the silkworm leads to spermatogenesis defects but does not affect sexual development [56]. The *poly(A)-specific ribonuclease-like domain-containing* (*Pnldc1*) is necessary for piRNA maturation in silkworms, and mutation of *Pnldc1* leads to abnormalities in nuclei of cells in eupyrene sperm bundles but not cells of other organs [57,58]. We have generated transgenic lines using CRISPR-Cas9 technology with mutations in *Zucchini* and *Papi*, which encode enzymes required for 3’-end processing of piRNAs, in *Tdrd12* and *Tudor-SN*, which are involved in ping-pong cycling, and in *Yu*, *Armi* and *Mino*, which encode chaperonins that function during piRNAs biogenesis, but none of these lines exhibit any obvious phenotypes (unpublished data). However, all these genes are crucial for piRNA processing in *Drosophila,* and their mutation will cause sterility, whereas our results indicate that the piRNA processing pathway is also different between silkworms and flies, and the processing of the primary signal *Fem*-piRNA is independent of those elements [52]. Furthermore, we do not understand how *Fem* transcription is activated, and it is possible that piRNAs other than *Fem* or non-coding RNAs from W are involved in sex determination. If other non-coding RNAs from W are involved in the female sex determination of *B. mori*, their role is expected to be minor if compared with Fem. Indeed, *Fem* repression is sufficient to cause masculinization of *dsx* splicing in embryos.

## 4. Masculinizers, the Possible Transducers 

*Masc* is the target gene of the primary signal mediated by *Fem* [24]. Evolutionary analysis revealed that *Masc* is conserved among the species in Lepidoptera but is not found in other insects, indicating that the sex determination pathway in Lepidoptera is likely distinct [18]. In the silkworm, *Masc* is located on the Z chromosome [24]. It encodes a CCCH-tandem zinc-finger protein and controls both masculinization and dosage compensation [59]. Recent studies have demonstrated that *Masc* is also required for masculinization in multiple lepidopterans including *Trilocha varians*, *Ostrinia furnacalis* and *Agrotis ipsilon* [21,34,35]. Besides, it is interesting that in *O. furnacalis*, *Wolbachia*-induced male-specific lethality is also caused by a failure of dosage compensation via suppression of *Masc* [60]. However, it remains to be determined whether Masc functions as masculinizer among all the species in the Lepidoptera order. A novel splice variant of *Masc* (*Masc-S*), which lacks the intact sequence of the cleavage site for *Fem*-piRNA, encodes a C-terminal truncated protein that has been identified in both sexes. The variant of *Masc*, *Masc-S*, participates in female genital development in the silkworm [61]. Moreover, there is still a gap between Masc and *dsx*, and further investigations that identify the direct targets of Masc will be able to clarify its role. 

In *Drosophila* females, female-specific *dsx* is promoted by Tra and Tra-2 as trans-acting splicing activators [5,29] (Figure 2E). In *Bombyx*, lacking *tra* ortholog, the sex-specific splicing regulation of the *dsx* gene is different (Figure 2F). A splicing factor might have replaced the function as Tra in *transformer*-less species during evolution [9,62,63]. PSI is a KH-domain RNA-binding splicing factor that regulates tissue-specific alternative splicing of P-element transposon transcripts to restrict transposition activity to germ-line tissues and that influences development and mating behavior in flies [64,65,66,67,68,69]. Though PSI is not involved in the sex determination pathway in *Drosophila*, it is a key auxiliary factor in silkworm male sex determination. PSI binds to exon 4 of *dsx* pre-mRNA and facilitates its male-specific splicing [36]. Genetic evidence, by using a transgenic CRISPR/Cas9 system, has shown that the mutation of *PSI* will cause partial male-to-female defects, whereas it has been demonstrated that it is indispensable for masculinization in the silkworm [33]. It is still unclear that *PSI* affects only males, however, as it is present in both sexes. Besides, according to published reference genomes, *PSI* is present in multiple lepidopterans, but whether it is involved in sex determination still remains to be identified [70]. Recently, it was reported that a *Znf-2* mutant has a phenotype similar to that of the *PSI* loss-of-function mutant [27]. *Znf-2* encodes a CCCH-type zinc finger protein and contributes to male-specific splicing of *dsx* in silkworm cell lines [71]. The *Znf-2* mutant males have feminized external genitalia, and the female isoform of *dsx* is detected in males, indicating that *Znf-2* is essential for male sexual differentiation [27]. In addition, larval development is delayed and body size diminished in *PSI* and *Znf-2* mutant males, demonstrating that these factors have functions in development in the silkworm. 

There is strong evidence supporting that Masc, PSI and Znf-2 are indispensable for masculinization, but where and how exactly these factors function in the sex determination cascade is not known. These questions may be answered by dissection of the functional relationships among Masc, PSI and Znf-2, and by identification of additional players in the sex determination pathway. Recent studies have identified components as BxRBP1, BxRBP2 and BxRBP3 (*Bombyx mori dsx* RNA-binding proteins) are *dsx* pre-mRNA binding protein by using the strategy of yeast three-hybrid screening. It is interesting that the alternatively transcribed *BxRBP3* isoforms are regulated by Masc and physically interact with PSI, but whether *BxRBP3* is the transducer remains to be verified via genetic methods [72] (Figure 2D,F). However, based on the observation that females can be completely reversed to males when the signal transducer *tra* is knocked-down or knocked-out in Diptera, we speculate that the transducer in the silkworm either has not yet been identified or may act very differently in the sex determination cascade [19,73,74,75,76]. 

## 5. Doublesex, the Executor

DSX protein sex-specific isoforms are conserved in insects and act as the downstream regulator in the sex determination pathway [31]. DSX controls sexual dimorphic development in insects, and mutation of sex-specific isoforms will cause male or female reproductive defects, respectively; hence, it has been targeted in development of sterile insect biotechnology [77,78,79,80]. As in Diptera, also in Lepidoptera, the *dsx* pre-mRNA is alternatively spliced to encode transcription factors that contain a common N-terminal domain and a sex-specific C-terminal domain [81]. The N-terminal domain possesses a DNA-binding motif that mediates the binding of DSX to target genes, whereas the C-terminal domain is crucial for DSX regulation of sexual development [37,82]. Disruptions of functions of certain isoforms of DSX lead to either sex-specific sexual-dimorphic defects or intersexual phenotypes in multiple Lepidoptera insects including *B. mori*, *A. ipsilon*, *Plutella xylostella* and *H. cunea* [38,83,84,85]. 

In *Drosophila*, the transcription cofactor Intersex (Ix) functions together with the female-specific product of *dsx*, DSXF, to implement female sexual differentiation [86] (Figure 2E). However, depletion of *ix* in the silkworm does not affect gonad development or splicing of the *dsx* pre-mRNA in either sex, suggesting that *ix* functions differently in silkworms than in flies [87]. The transcription factor Fruitless (Fru) plays roles in sexual behaviors and is regulated by the sex determination cascade in flies [7]. A recent study in the silkworm demonstrated that Fru is also affected by the upstream sex-determining factors and acts together with DSX to regulate both mating and courtship behavior [88]. 

Although DSX exists in the majority of insect species, the mechanisms by which it functions in sex determination and subsequent developmental processes are poorly understood in most insect species. Only a few genes directly controlled by DSX have been identified, although numerous targets of DSX have been predicted and analyzed on the genome-wide scale in *Drosophila* [89,90]. It remains to be determined whether the events downstream of DSX were conserved among insects during evolution [91].

## 6. Conclusions and Perspectives

Genetic, molecular and biochemical studies demonstrate that the sex-determination system in the silkworm, a model lepidopteran insect, is very different from that in Diptera [11]. Although several factors have been characterized in the silkworm, how they are functionally correlated with each other is still unclear [24,25,26,27,28,33]. Moreover, no factors that specifically control sex transformation have been discovered, making the sex determination cascade in the silkworm elusive. If *Fem* as the primary signal is dominant in the sex determination cascade, overexpressing *Fem* in males may cause a male-to-female change. However, based on previous studies and our work, we postulate that there may exist an F-factor other than *Fem* as the primary signal. Besides, although SIWI and Gtsf1 have been shown to be piRNA-binding proteins, there is no direct in vivo evidence that these proteins bind to *Fem*. RNA-binding protein immunoprecipitation assays are required to verify this binding. 

It will also be worth further investigating Masc-interacting components and downstream targets in future studies. For instance, comparison of the epigenetic markers involved in histone modifications in *Masc* mutants of both sexes may help us understand the dosage compensation mechanism in the silkworm. Moreover, in *transformer*-less species, the alternative splicing regulation of *dsx* remains unclear. Since PSI exists in both sexes in the silkworm, we speculate that the masculinization function of PSI may depend on Masc [33,36]. No evidence has demonstrated that there is a physical interaction between Masc and PSI. Although components have been identified as PSI-interacting proteins, the expressing patterns among them have no sex bias [97,98]. Based on regulation by Masc, and direct interaction with PSI and *dsx* pre-mRNA, it seems BxRBP3 has the potential function as the transducer [72]. Furthermore, *Masc* is also involved in masculinization in multiple lepidopterans, and whether a general cascade is conserved among them requires investigation [21,34,35]. Therefore, further research on the mechanism of Masc will likely be the key to solving this puzzle.

The Lepidoptera order includes both pest species and species of economic importance. Thus, it is important to understand the general mechanism of sex determination in lepidopterans. Here we compared the differences in key factors involved in sex determination in the Lepidoptera model insect silkworm and *Drosophila,* and we proposed some future research directions. However, it should be noted that the conception “primary signal-transducer-executor” of sex determination cascade may not fit in all insects. Perhaps in mosquitos, the primary signal gene, as a splicing factor, might directly regulate the alternative splicing of *dsx* [12,13]. With further application of next-generation sequencing and of genetic research tools in non-model insects, the diverse cascades in insects will be uncovered. By targeting sex determination genes, it will guide development of techniques for pest control. It will also further our understanding of how evolutionary selection in insects leads to adaptation and natural selection, which should in turn enable selection or engineering of desired traits in species of economic importance [99]. 

## Figures and Tables

**Figure 1 genes-12-00315-f001:**
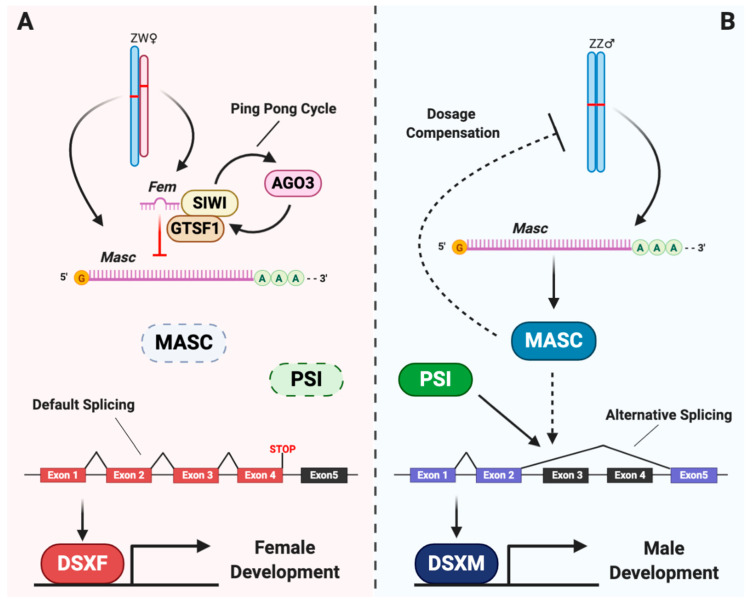
The preliminary established genetic cascade of sex determination in *Bombyx mori*. Female development is the default sex determination pathway in *B. mori* (**A**). The *Feminizer* piRNA (*Fem*) derived from the female W chromosome interacts with the proteins *Silkworm-PIWI* (SIWI) and *gametocyte-specific factor 1* (GTSF1) to form as piRNA-induced silencing complexes and silence *Masculinizer* (*Masc*) at the mRNA level during ping-pong amplification with Argonaute RISC Catalytic Component 3 (AGO3) [24,25]. The absence of *Masc* protein (MASC) and an inactive *P-element Somatic Inhibitor* protein (PSI) results in production of the female-specific isoform of *dsx* (*dsxF*), which contain all of the exons of the gene [33]. The female-specific DSX (DSXF) results in female morphological sex characteristics [37,38]. MASC is expressed normally in the male due to the lack of *Fem* and presence of active PSI (**B**), and this results in dosage compensation and splicing that produces the male-specific isoform of *dsx* (*dsxM*), which contains exons of 1, 2, and 5 of the *dsx* pre-mRNA [33,36]. The male-specific DSX (DSXM) promotes male development [37,38].

**Figure 2 genes-12-00315-f002:**
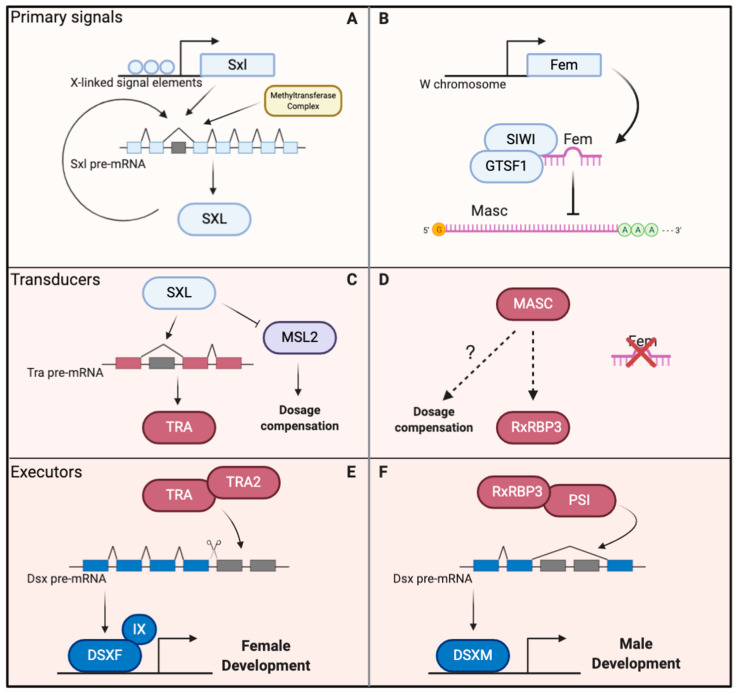
Comparison of the sex determination cascade between *Drosophila melanogaster* (**A**,**C**,**E**) and *Bombyx mori* (**B**,**D**,**F**). In female *D. melanogaster*, the dosage of several X-linked signal element proteins regulates the transcription initialization of *Sex-lethal* (*Sxl*). An autoregulatory feedback loop ensures continuous expression. Methyltransferase complex components also regulate the precise splicing of *Sxl* (**A**) [92,93,94,95]. The *Sxl* protein (SXL) directly binds to *transformer* (*tra*) pre-mRNA and regulates its alternative splicing. *Male-specific-lethal 2 (Msl2)* is inhibited at the mRNA level (**C**). Functional *transformer* protein (TRA) with *transformer-2* protein (TRA2) together regulate *doublesex* (*dsx*) female-specific alternative splicing (**E**) [5]. The female-specific DSX (DSXF) interacts with *Intersex* protein (IX) to promote female development [86]. In female *B. mori*, the *Fem*inizer piRNA (*Fem*) derived from the female W chromosome is postulated to interact with the protein *Silkworm-PIWI* (SIWI) and *gametocyte-specific factor 1* (GTSF1) to form the piRNA-induced silencing complexes that silences *Masculinizer* (*Masc*) at the mRNA level (**B**) [24,25,28]. In male *B. mori*, the *Masc* protein (MASC) increases the expression of *Bombyx mori dsx RNA-binding protein 3* (RxRBP3) (**D**). RxRBP3, together with the protein of *P-element somatic inhibitor* (PSI), regulates the male-specific alternative splicing of *dsx* (**F**) [26,72,87,96]. The male-specific DSX (DSXM) promotes male development [37,38].

## Data Availability

No new data were created or analyzed in this study. Data sharing is not applicable to this article.

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
