# Peer review of "The Sex Determination Cascade in the Silkworm"

_genes, 2021, doi:10.3390/genes12020315_

Round 1

Reviewer 1 Report

Manuscript #: genes-1023377

Title: "The sex determination cascade in lepidopteran insects"

Authors: Yang et al.,

Comments

This is a well-written review that explains the latest findings on the sex-determining mechanism of silkworms and other lepidopteran insects. However, I think it is necessary to make corrections by referring to the comments below before publishing.

Major comments for revision

1. Lines 42-44. I think this sentence is incorrect because Nix has been identified from Aedes aegypti, which is one of dipteran species that do not have a tra ortholog in their genome. Thus, "the primary signals mediated by these genes are all transduced by the ortholog of Tra" is incorrect. Please refer to the following papers.

Geuverink E, Beukeboom LW. (2014) Phylogenetic distribution and evolutionary dynamics of the sex determination genes doublesex and transformer in insects. Sex Dev. 8:38-49.

Salvemini M, D'Amato R, Petrella V, Aceto S, Nimmo D, Neira M, Alphey L, Polito LC, Saccone G. (2013) The orthologue of the fruitfly sex behaviour gene fruitless in the mosquito Aedes aegypti: evolution of genomic organisation and alternative  splicing. PLoS One. 8:e48554.

Salvemini M, Mauro U, Lombardo F, Milano A, Zazzaro V, Arcà B, Polito LC, Saccone G. (2011) Genomic organization and splicing evolution of the doublesex gene, a Drosophila regulator of sexual differentiation, in the dengue and yellow fever mosquito Aedes aegypti. BMC Evol Biol. 11:41.

2. Lines 62-64. I think this sentence is also incorrect because of the reason described above. Some dipteran species such as A. aegypti, Anopheles gambiae, and Culex quinquefasciatus do not have a tra ortholog in their genome. Please refer to the following paper.

Geuverink E, Beukeboom LW. (2014) Phylogenetic distribution and evolutionary dynamics of the sex determination genes doublesex and transformer in insects. Sex Dev. 8:38-49.

3. Lines 130-131. The authors said that Masc functions in males only. But the resent study described by Qin et al. (2019) demonstrated that Masc is important for the development of the female external genitalia. I think the authors should correct the sentence based on the following paper.

Qin Z, Juan L , Mao W, He W, Yao W, Kai W, Qiu W,  Xing Z.(2019) A Novel Splice Variant of the Masculinizing Gene Masc with piRNA-Cleavage-Site Defect Functions in Female External Genital Development in the Silkworm, Bombyx mori. Biomolecules, 9(8):318.

Author Response

Reviewer:1

  1. Lines 42-44. I think this sentence is incorrect because Nix has been identified from Aedes aegypti, which is one of dipteran species that do not have a tra ortholog in their genome. Thus, "the primary signals mediated by these genes are all transduced by the ortholog of Tra" is incorrect.

RESPONSE: We correct our mistake as ‘Although these factors are diverse, the primary signal mediated by these genes are all transduced by a pathway that leads to alternative splicing of Dsx, which controls sex differentiation eventually.’ (Lines 43-45)

  1. Lines 62-64. I think this sentence is also incorrect because of the reason described above. Some dipteran species such as A. aegypti, Anopheles gambiae, and Culex quinquefasciatus do not have a tra ortholog in their genome.

RESPONSE: We correct our mistake as ‘’In Drosophila melanogasler, Musca dometica, Ceratitis captitata and Nasonia vitripennis the primary sex determiner controls sex development via Tra-transduced signaling pathway, resulting in the alternative splicing of Dsx.’(Lines 65-68)

  1. Lines 130-131. The authors said that Masc functions in males only. But the resent study described by Qin et al. (2019) demonstrated that Masc is important for the development of the female external genitalia. I think the authors should correct the sentence based on the following paper.

RESPONSE: We deleted lines 130-131“Masc functions in males only” and add the citation which the reviewer mentioned at line 136.

Reviewer 2 Report

This review paper by Yang et al. summarizes what is known about the molecular mechanisms of sex determination in Bombyx mori, a model lepidopteran insect.

I think the work does a satisfactory job of describing the current state of knowledge of sexual development in holometabolic insects (such as Drosophila, Tribolium, and Bombyx), but there are a few weaknesses of the review. First and foremost, it isn’t clear what new knowledge is brought by the review. The figures don’t really help to clear this up, and I’m not sure what message I’m supposed to take from this paper.

One of the strengths of the paper is the organization style of Signal-Transducer-Executor, but again it’s not clear whether this nomenclature is a new idea, or if it is a stylistic choice.

I think one thing that would help would be to more clearly explain and summarize the differences between lepidopterans and other holometabolic insects. It’s also not clear what insights B. mori brings to the discussion of transformer-less sex determination mechanisms. Lepidopterans are not unique in lacking transformer. For example, order Strepsiptera, order Pthiraptera, and order Hemiptera (two of which aren’t even holometabolic orders!) all lack transformer (per Geuverink and Beukeboom (2014) Phylogenetic Distribution and Evolutionary Dynamics of the Sex Determination Genes doublesex and transformer in Insects). What advantage or insights are gained in B. mori about this system?

I think something that would help would be either a figure or a table directly comparing the elements of a transformer-less sex determination cascade to the “typical” transformer pathway in Drosophila.

I don’t think it’s accurate to say that “Doublesex is conserved across all animals” (line 162). I think it needs more explanation since Dsx isn’t conserved, and seems to be insect-specific. Instead, it should be clarified that Dsx and other DM-binding genes are conserved, and the DM binding-region should be mentioned earlier to help clarify this point.

The “ping-pong” cycle needs to be explained, at least briefly, since it’s mentioned several times but then disregarded.

I think overall the stated goal of the authors on line 56 is “Here we summarize what is known about lepidopteran sex determination and suggest that further studies in the Lepidoptera model Bombyx mori will provide insights into sex determination in Lepidoptera and a better understanding of the diversity of sex determination in different species.” The authors definitely summarize sex determination and suggest that more studies can provide insight into Lepidoptera, but I’m not sure how this information will contribute to understanding species diversity in sex determination, since B. mori is a well-studied (and derived) laboratory model. Perhaps mentioning experiments that could be more generalized across lepidoptera, or comparing what is known about B. mori to, say, Manduca would help accomplish this goal.

In addition to the lack of a central thesis or message, there are a few editing errors, mostly missing linking words such as “is” or getting is/are confused when referring to lists or singular items. For examples, on lines 34-35, the plural is confusing as the list moves on and on. Perhaps splitting this sentence into two sentences would be more clear. On line 46, instead of saying “a diverse order OF insects”, the authors just write “a diverse order insects”. Again, small errors, but they persist through the manuscript.

I think ultimately the manuscript needs a bit of revision to make the central message more clear. Right now the only thing I leave this manuscript thinking is mentioned on line 200 “RNA-binding  protein  immunoprecipitation assays are required  to  verify  this binding”. Is this the central takeaway from the paper? If not, what is the message of this review?

My suggestions are as follows- Decide the central thesis of the paper and make it more clear in the introduction. Add more explicit mention of transformer-less mechanisms in other insects. Add more comparison to other model lepidopterans. Add a more visual comparison of what is known between Drosophila and B. mori. Fix the editing errors.

It would also maybe be an interesting fix to more clearly identify what recent experiments have moved the field forward in understanding this problem, perhaps by adding citations to mechanisms in Figure 1.

Author Response

Reviewer:2

  1. I think the work does a satisfactory job of describing the current state of knowledge of sexual development in holometabolic insects (such asDrosophila,Tribolium, and Bombyx), but there are a few weaknesses of the review. First and foremost, it isn’t clear what new knowledge is brought by the review. The figures don’t really help to clear this up, and I’m not sure what message I’m supposed to take from this paper.

RESPONSE: Our central message is indicating that the sex determination pathway in silkworm is still ambiguous and we have discussed new viewpoints at each different section that will benefit to researcher which are focusing on sex determination in silkworms and other lepidopterans. Indeed, we are not sure that what message can researchers who are focusing on other insects could take. (lines 56-58)

The reasons we consider our review will benefit researchers in the field of lepidopterans as below.

Because till now there is no complete reference genome of the W chromosome due to replicate sequences, it is still unknown that whether Fem is the only F-factor from the W. Besides we consider understanding the mechanism of Fem transcription initial, in another word, understanding the mechanism of the primary signal transcription initial is crucial for researchers. Besides, we also did a lot of work on piRNAs biogenesis for the purpose of understanding the process of Fem and expect Siwi and Gtsf1 is involved in this process, no other factors were verified so far. I think experts in the field of piRNAs did not care to much about sex determination as researchers in the field of sexual development did not pay much attention to piRNAs. But again, due to the special function of piRNA Fem, it is important to fully understand the piRNAs mechanism, especially by a genetic way. (lines 129-131)

In the other hand, we consider that understand the mechanism of Masc is the key to the whole picture. Although researchers have discovered that Masc is not only involved in sex determination but will also affect the dosage compensation, how Masc participates in these functions is still elusive. We believe there is a transducer like transformer, it could be downstream of Masc and might directly interact with PSI. (lines 159-165)

  1. I think one thing that would help would be to more clearly explain and summarize the differences between lepidopterans and other holometabolic insects. It’s also not clear what insightsB. moribrings to the discussion oftransformer-less sex determination mechanisms. Lepidopterans are not unique in lacking transformer. For example, order Strepsiptera, order Pthiraptera, and order Hemiptera (two of which aren’t even holometabolic orders!) all lack transformer. What advantage or insights are gained in B. mori about this system?

RESPONSE: We still know little about sex determination in other lepidopterans. All we know is that Masc function as masculinizers in Trilocha varians, Ostrinia furnacalis and Agrotis ipsilon, and Dsx is crucial for sexual development in Agrotis ipsilon, Plutella xylostella and Hyphantria cunea. We do not know whether factors as Siwi, Gtsf1, PSI and znf-2 is also involved in sex determination cascade of these species. Due to the limit knowledge, we cannot tell what general mechanism is shared among transformer-less species. (lines 137-140 and lines 151-153)

  1. I think something that would help would be either a figure or a table directly comparing the elements of atransformer-less sex determination cascade to the “typical”transformer pathway in Drosophila.

RESPONSE: We also add a table which summarizes the key components as ‘primary signal’ and ‘transducer’ of different species mentioned in our review.

  1. I don’t think it’s accurate to say that “Doublesex is conserved across all animals” (line 162). I think it needs more explanation since Dsx isn’t conserved, and seems to be insect-specific. Instead, it should be clarified that Dsx and other DM-binding genes are conserved, and the DM binding-region should be mentioned earlier to help clarify this point.

RESPONSE: We correct the sentence as ‘DSX and DM domain genes are conserved in animals and acts as the downstream regulator in the sex determination pathway.’(lines 167-168)

  1. My suggestions are as follows- Decide the central thesis of the paper and make it more clear in the introduction. Add more explicit mention of transformer-less mechanisms in other insects. Add more comparison to other model lepidopterans. Add a more visual comparison of what is known between Drosophilaand B. mori. Fix the editing errors.

RESPONSE: We have revised our central thesis in the part of the introduction as ‘Here we summarize what is known about lepidopteran sex determination and suggest that further studies in the Lepidoptera model Bombyx mori will provide insights into sex determination in other Lepidoptera species.’ (lines 56-58)

We have added more explicit mention of transformer-less mechanisms in other insects (lines 45- 46 and lines71-73).

We also add a table which summarizes the key components as ‘primary signal’ and ‘transducer’ of different species mentioned in our review.

Reviewer 3 Report

This review deals with a hot topic. In Lepidoptera, the key components of the sex determination pathway have only recently been discovered in a model species, the silkworm Bombyx mori, by a Japanese group led by Susumu Katsuma (Kiuchi et al. 2014). Since then, the group of Yongping Huang, i.e. authors of this review, is among a few groups, who significantly contributed to the understanding of this pathway, especially by identifying other genes involved in the pathway, such as Gtsf1. Their contribution thus gives them full credit to write this review.

The review is well organized. First, the authors describe general principles of sex determination in insects and summarize the current knowledge on the sex-determining cascade in the silkworm, along with few findings in other lepidopteran species. Then, they focus in three subchapters on three main components of sex determination in the silkworm, which are in fact similar to general components in all insects: feminizers, masculinizers (transducers), and executors. In the last subchapter, they raise a number of unanswered questions and indicate the direction of further research.

The review is clearly written and cites the most relevant articles. However, the authors are not patient enough to follow genetic rules for writing gene symbols. As this review can be expected to be followed by many readers, it is very important to follow these rules so that the article does not bring chaos to the genetic nomenclature. The authors were also not patient in formatting References. Overall, this review is significant and deserves publication after revising a number of inaccuracies listed below.

Specific comments

1.Figure 1 and legend: at the top of Fig. 1A and Fig. 1B, each sex chromosome is shown as composed of two sister chromatids and the females-specific chromosome W is shown in pink, whereas Z chromosomes in blue, which is indicative. Nevertheless, for readers less familiar with the sex chromosome system in the silkworm, it would be better to add also letters “Z” and “W” to make it clear. In addition, “AGO3” in panel A should be explained in the figure legend.

2. References 50 and 51, though both very important, are not cited in the text!

3. Figure 2: in this figure (and also in Fig. 1), authors illustrated the Bombyx W chromosome in panel A as much smaller than the Z chromosome. However, in Bombyx as well as in the majority of Lepidoptera, the W chromosome is usually of similar size as the Z chromosome, or only slightly smaller. Therefore, I recommend illustrating the W chromosome as similar in size to the Z chromosome in both Figures.

Minor suggestions

Line 12 and throughout the text: I believe that the correct spelling of “executers” is “executors”. Please check.

Line 29: correct “miosis” to “meiosis”

Line 32: correct the following symbols of chromosomes “X0” and “Z0”; they should be written with “0” as “zero” but not with “O”. The zero indicates the absence of the second sex chromosome.

Line 35 and further in the text: “tra” as a symbol for transformer gene should be written with a small initial letter (see FlyBase)

Line 35 and further in the text: doublesex (dsx) should be written with a small initial letter (see FlyBase)

Line 40: Mdmd should be written in italics

Line 41: Nix should be written in italics

Line 41: tra2 should be written in italics and with a small initial letter

Line 42: MoY should be written in italics

Line 43: tra should be written in italics and with a small initial letter

Line 46: diverse order of insects

Line 49/50 and further in the text: I prefer WZ to ZW symbols for females because “WZ” order of the symbols was originally established by T.H. Morgan and his colleagues in 1915 and it is also used in all main papers and review articles on sex chromosomes in Lepidoptera.

Line 51: tra is a gene symbol here and should be written with a small initial letter

Line 68: tra should be written with a small initial letter

Line 137: Wolbachia should be written in italics

Line 158: females can be completely reversed to males

Line 158: tra should be written with a small initial letter

Line 172: “ix” as a symbol for intersex gene should be written with a small initial letter (see FlyBase)

General comment to References from line 227 onwards: all title words, except the first word and scientific and geographic names, should be written with a small initial letter.

Line 232: Chromosome Res.

Line236: Drosophila should be written in italics

Line 238: “csd” is a gene name and should be written in italics

Line 250: Aedes aegypti should be written in italics

Line 253/254: Nix, Aedes aegypti, and myo-sex should be written in italics

Line 257/258: Mdmd and CWC22 should be written in italics

Line 260: Maleness-on-the-Y and MoY should be written in italics

Line 264: Masc should be written in italics

Line 265: Agrotis ipsilon should be written in italics

Line 269: Bombyx mori should be written in italics

Line 272: correct publication year in the article of Traut et al. to 2007 [explanation: the printed version of this article was published in 2007; then, the online version was additionally released in 2008]

Line 277: Bombyx mori and BmAsh2 should be written in italics

Line 281: Masculinizer should be written in italics

Line 282: Bombyx mori should be written in italics

Line 284: Bmznf-2 should be written in italics

Line 286: Gtsf1 should be written in italics

Line 287: Bombyx mori should be written in italics

Line 289: transformer should be written in italics

Line 295: Bombyx mori should be written in italics

Line 298: Sex-lethal should be written in italics

Line 301: Bombyx mori and BmPSI should be written in italics

Line 305: Masculinizer and Trilocha varians should be written in italics

Line 308: Ostrinia furnacalis and Masculinizer should be written in italics

Line 312: Bombyx mori and dsx should be written in italics

Line 315: Bombyx mori should be written in italics

Line 317: Bombyx mori and doublesex should be written in italics

Line 320: Bombyx mori should be written in italics

Line 323: Bombyx mori should be written in italics

Line 326: Bombyx mori should be written in italics

Line 327: Bombyx should be written in italics

Line 344: Drosophila should be written in italics

Line 348: Maelstrom is a protein name and should be written with capital M

Line 351: Bombyx mori should be written in italics

Line 359: Bombyx mori should be written in italics

Line 362: Wolbachia should be written in italics

Line 364: Drosophila melanogaster should be written in italics

Line 375: correct journal name “Rna” to “RNA”

Line 378: Drosophila should be written in italics

Line 383: doublesex and Anopheles gambiae should be written in italics

Line 386: Anopheles gambiae should be written in italics

Line 389: Bombyx mori should be written in italics

Line 392: Bombyx mori should be written in italics

Line 394: doublesex should be written in italics

Line 395: Drosophila melanogaster and Bombyx mori should be written in italics

Line 398: doublesex and Bombyx mori should be written in italics   

Line 399: Intersex should be written in italics

Line 402: sex-determination

Line 404: D. melanogaster should be written in italics

Line 407: Drosophila should be written with capital “D” and in italics; Doublesex is a protein in the case and should be written with capital “D”

Line 410: doublesex should be written in italics

Author Response

Reviewer 3

1.Figure 1 and legend: at the top of Fig. 1A and Fig. 1B, each sex chromosome is shown as composed of two sister chromatids and the females-specific chromosome W is shown in pink, whereas Z chromosomes in blue, which is indicative. Nevertheless, for readers less familiar with the sex chromosome system in the silkworm, it would be better to add also letters “Z” and “W” to make it clear. In addition, “AGO3” in panel A should be explained in the figure legend.

RESPONSE: We adjust our figure as the reviewer suggested. (Figure1)

  1. References 50 and 51, though both very important, are not cited in the text!

RESPONSE: We add the Reference at line 137.

  1. Figure 2: in this figure (and also in Fig. 1), authors illustrated theBombyxW chromosome in panel A as much smaller than the Z chromosome. However, in Bombyx as well as in the majority of Lepidoptera, the W chromosome is usually of similar size as the Z chromosome, or only slightly smaller. Therefore, I recommend illustrating the W chromosome as similar in size to the Z chromosome in both Figures.

RESPONSE: We adjust our figure as the reviewer suggested. (Figure2)

Minor suggestions

Line 12 and throughout the text: I believe that the correct spelling of “executers” is “executors”. Please check.

Line 29: correct “miosis” to “meiosis”

Line 32: correct the following symbols of chromosomes “X0” and “Z0”; they should be written with “0” as “zero” but not with “O”. The zero indicates the absence of the second sex chromosome.

Line 35 and further in the text: “tra” as a symbol for transformer gene should be written with a small initial letter (see FlyBase)

Line 35 and further in the text: doublesex (dsx) should be written with a small initial letter (see FlyBase)

Line 40: Mdmd should be written in italics

Line 41: Nix should be written in italics

Line 41: tra2 should be written in italics and with a small initial letter

Line 42: MoY should be written in italics

Line 43: tra should be written in italics and with a small initial letter

Line 46: diverse order of insects

Line 49/50 and further in the text: I prefer WZ to ZW symbols for females because “WZ” order of the symbols was originally established by T.H. Morgan and his colleagues in 1915 and it is also used in all main papers and review articles on sex chromosomes in Lepidoptera.

Line 51: tra is a gene symbol here and should be written with a small initial letter

Line 68: tra should be written with a small initial letter

Line 137: Wolbachia should be written in italics

Line 158: females can be completely reversed to males

Line 158: tra should be written with a small initial letter

Line 172: “ix” as a symbol for intersex gene should be written with a small initial letter (see FlyBase)

General comment to References from line 227 onwards: all title words, except the first word and scientific and geographic names, should be written with a small initial letter.

Line 232: Chromosome Res.

Line236: Drosophila should be written in italics

Line 238: “csd” is a gene name and should be written in italics

Line 250: Aedes aegypti should be written in italics

Line 253/254: Nix, Aedes aegypti, and myo-sex should be written in italics

Line 257/258: Mdmd and CWC22 should be written in italics

Line 260: Maleness-on-the-Y and MoY should be written in italics

Line 264: Masc should be written in italics

Line 265: Agrotis ipsilon should be written in italics

Line 269: Bombyx mori should be written in italics

Line 272: correct publication year in the article of Traut et al. to 2007 [explanation: the printed version of this article was published in 2007; then, the online version was additionally released in 2008]

Line 277: Bombyx mori and BmAsh2 should be written in italics

Line 281: Masculinizer should be written in italics

Line 282: Bombyx mori should be written in italics

Line 284: Bmznf-2 should be written in italics

Line 286: Gtsf1 should be written in italics

Line 287: Bombyx mori should be written in italics

Line 289: transformer should be written in italics

Line 295: Bombyx mori should be written in italics

Line 298: Sex-lethal should be written in italics

Line 301: Bombyx mori and BmPSI should be written in italics

Line 305: Masculinizer and Trilocha varians should be written in italics

Line 308: Ostrinia furnacalis and Masculinizer should be written in italics

Line 312: Bombyx mori and dsx should be written in italics

Line 315: Bombyx mori should be written in italics

Line 317: Bombyx mori and doublesex should be written in italics

Line 320: Bombyx mori should be written in italics

Line 323: Bombyx mori should be written in italics

Line 326: Bombyx mori should be written in italics

Line 327: Bombyx should be written in italics

Line 344: Drosophila should be written in italics

Line 348: Maelstrom is a protein name and should be written with capital M

Line 351: Bombyx mori should be written in italics

Line 359: Bombyx mori should be written in italics

Line 362: Wolbachia should be written in italics

Line 364: Drosophila melanogaster should be written in italics

Line 375: correct journal name “Rna” to “RNA”

Line 378: Drosophila should be written in italics

Line 383: doublesex and Anopheles gambiae should be written in italics

Line 386: Anopheles gambiae should be written in italics

Line 389: Bombyx mori should be written in italics

Line 392: Bombyx mori should be written in italics

Line 394: doublesex should be written in italics

Line 395: Drosophila melanogaster and Bombyx mori should be written in italics

Line 398: doublesex and Bombyx mori should be written in italics   

Line 399: Intersex should be written in italics

Line 402: sex-determination

Line 404: D. melanogaster should be written in italics

Line 407: Drosophila should be written with capital “D” and in italics; Doublesex is a protein in the case and should be written with capital “D”

Line 410: doublesex should be written in italics

RESPONSE: We have corrected all the mistakes the reviewer have mentioned as minor suggestions.

Round 2

Reviewer 1 Report

The revised manuscript is now suitable for publication. I am satisfied with the author's responses and plausible explanations.

Author Response

Reviewer 1

Thank you very much for your review report.

Reviewer 2 Report

These changes do not really address many of my concerns. I still don't understand what the point of this review is. It seems this paper has already been written by Katsuma et al (2018; https://www.ncbi.nlm.nih.gov/pmc/articles/PMC6021594/). I don't understand what new insights this current review brings to the field. Moreover, many of the changes don't address my actual comments.

The authors acknowledge that their review is incredibly limited in scope several times in their response, which leads me to ask whether or not this article is suitable for this journal. Would it not be better served in an insect-specific journal, since it has been written (especially after the edits) to apply only to one specific order of insects?

In their response the authors say "Besides, we also did a lot of work on piRNAs biogenesis for the purpose of understanding the process of Fem and expect Siwi and Gtsf1 is involved in this process, no other factors were verified so far. I think experts in the field of piRNAs did not care to much about sex determination as researchers in the field of sexual development did not pay much attention to piRNAs."

My response is this- is the current review supposed to be written to appeal to piRNA experts? If so, I don't think the current manuscript is appealing in that realm. On the other hand, if the review is arguing that sexual development researchers should pay more attention to piRNAs, then the review also doesn't accomplish that goal.

In my first review I mentioned that "I think something that would help would be either a figure or a table directly comparing the elements of a transformer-less sex determination cascade to the “typical” transformer pathway in Drosophila."

The authors response was to list a table containing the signals and transducers in a bunch of other insects, which is not what I was recommending to add. If this is their goal, why leave out the one strength of the paper, which was the "Signal-Transducer-Executor" framework of the rest of the manuscript?

There are also several author comments that were not addressed. What is "ping-pong cycling"? Again, in comparison to Katsuma et al (2018), this manuscript is incredibly confusing. In Katsuma et al. the ping-pong cycle is explained even if not centrally important to the reader.

In summary, I don't feel the authors have addressed my concerns. They have revised their thesis barely at all (changing from "understanding the diversity of sex determination in different species" to "providing insights into sex determination in other Lepidopteran species"), yet the manuscript doesn't seem very clear about what insights they are trying to provide.

Again, my main question remains: What is the point of this Manuscript? Who is it written for, piRNA researchers or insect sex determination researchers?

Author Response

Reviewer2

These changes do not really address many of my concerns. I still don't understand what the point of this review is. It seems this paper has already been written by Katsuma et al (2018; https://www.ncbi.nlm.nih.gov/pmc/articles/PMC6021594/). I don't understand what new insights this current review brings to the field. Moreover, many of the changes don't address my actual comments. 

The authors acknowledge that their review is incredibly limited in scope several times in their response, which leads me to ask whether or not this article is suitable for this journal. Would it not be better served in an insect-specific journal, since it has been written (especially after the edits) to apply only to one specific order of insects?

RESPONESE: In the review written by Katsuma et al (2018; https://www.ncbi.nlm.nih.gov/pmc/articles/PMC6021594/), they summarized the discovery of Fem and the dosage compensation and masulinizatoin function of Masc, which we also mentioned from lines 91-111 and lines 133-141. They didn’t mention too much about the process of Fem piRNA (except ping-pong cycle).

  1. In our review, we mainly summarized our previous findings (ref 23, 26, 49, 51 and unpublished data). We have discovered that AGO3 is indispensable for Fem piRNA processing during ping-pong cycle (ref 23), we also discovered that components as Mael and Pnldc1 are crucial for piRNA processing, but no sex changes were observed in Mael and Pnldc1 mutants. We discovered that Gtsf1 is also involved in piRNA biogenesis which might affect Fem and caused partial female-to-male change eventually. However, the processing of Fem is still ambiguous. Besides, based on our understanding, we suggest there may exist an early F-factor other than Fem which dominants the sex determination cascade (lines 113-131 and 220-223).
  2. In this version, we also have summarized new findings as the variant of Masc-S, znf-2 and potential components as RxRBP3 involving in sex determination in Bombyx (lines 141-144, 153-167, 171-175, 231-235). We also put RxRBP3 into Figure2D and 2F.
  3. Although we organized the sex determination cascade as Signal-Transducer-Executor, we mentioned that this conception might not fit in all species (lines 235-238).
  4. Except the well-studied model Drosophila, only silkworm have identified several components which were involved in sex determination. In silkworms, not only by using RNA-sequencing for finding primary signals, but also have approached multiple strategies for searching key factors (ref 22, 34, 65, 86 and 87).

For example, in ref 65, cis-regulators of dsx were identified by using yeast-three hybrid. Perhaps this strategy is also useful for screening components which directly binds to dsx pre-mRNA in transformer-less species (lines 171-175).

In my first review I mentioned that "I think something that would help would be either a figure or a table directly comparing the elements of a transformer-less sex determination cascade to the “typical” transformer pathway in Drosophila."

The authors response was to list a table containing the signals and transducers in a bunch of other insects, which is not what I was recommending to add. If this is their goal, why leave out the one strength of the paper, which was the "Signal-Transducer-Executor" framework of the rest of the manuscript?

RESPONESE: We delete the table and the previous Figure 2 and replaces a figure which compares the key genes as “primary signals- transducers- executors” between Drosophila and Bombyx.

There are also several author comments that were not addressed. What is "ping-pong cycling"? Again, in comparison to Katsuma et al (2018), this manuscript is incredibly confusing. In Katsuma et al. the ping-pong cycle is explained even if not centrally important to the reader.

RESPONESE: We add a description at lines 106-109.

In summary, I don't feel the authors have addressed my concerns. They have revised their thesis barely at all (changing from "understanding the diversity of sex determination in different species" to "providing insights into sex determination in other Lepidopteran species"), yet the manuscript doesn't seem very clear about what insights they are trying to provide.

Again, my main question remains: What is the point of this Manuscript? Who is it written for, piRNA researchers or insect sex determination researchers?

RESPONESE: In this version, we add more explicit mention of transformer-less mechanisms (lines 66-67, 149-153 and 229-230). Expect fem in Apis mellifera, PSI in Bombyx mori and NIFmd in Nilaparvata lugens (preprint paper, under-revision), there seems no other reports about transformer-less mechanisms.

The paper is writing for researchers who are focusing on sex determination in lepidopterans. The reason why we put emphasis on piRNA is because Fem piRNA is the primary signal during sex determination in the silkworm (lines 100-131).

Reviewer3

Minor suggestions

Lines 19 and 36: doublesex

Lines 45, 70, 71, 92, 171, 181, 199: dsx

Line 74: … Lepidoptera, which indicates that the sex determination …

Line 87: dsx (dsxF)

Line 87: contains

Line 91: dsx (dsxM)

Line 330: correct publication year in the article of Traut et al. to 2007 [explanation: the printed version of this article was published in 2007; then, the online version was additionally released in 2008]

Lines 332-334: reference 36 is not cited in the revised text and should be deleted

RESPONESE: We correct all these mistakes.

Reviewer 3 Report

The authors solved satisfactorily all my specific comments and also my minor suggestions in the text. However, when reading the revised version of this review I found several other minor inaccuracies that need to be fixed. Nevertheless, the paper can be accepted and these minor errors can be fixed during further processing of the manuscript.

Minor suggestions

Lines 19 and 36: doublesex

Lines 45, 70, 71, 92, 171, 181, 199: dsx

Line 74: … Lepidoptera, which indicates that the sex determination …

Line 87: dsx (dsxF)

Line 87: contains

Line 91: dsx (dsxM)

Line 330: correct publication year in the article of Traut et al. to 2007 [explanation: the printed version of this article was published in 2007; then, the online version was additionally released in 2008]

Lines 332-334: reference 36 is not cited in the revised text and should be deleted

Author Response

Reviewer3

Minor suggestions

Lines 19 and 36: doublesex

Lines 45, 70, 71, 92, 171, 181, 199: dsx

Line 74: … Lepidoptera, which indicates that the sex determination …

Line 87: dsx (dsxF)

Line 87: contains

Line 91: dsx (dsxM)

Line 330: correct publication year in the article of Traut et al. to 2007 [explanation: the printed version of this article was published in 2007; then, the online version was additionally released in 2008]

Lines 332-334: reference 36 is not cited in the revised text and should be deleted

RESPONESE: We correct all these mistakes.